# Intelligent Identification and Features Attribution of Saline–Alkali-Tolerant Rice Varieties Based on Raman Spectroscopy

**DOI:** 10.3390/plants11091210

**Published:** 2022-04-29

**Authors:** Bo Ma, Chuanzeng Liu, Jifang Hu, Kai Liu, Fuyang Zhao, Junqiang Wang, Xin Zhao, Zhenhua Guo, Lijuan Song, Yongcai Lai, Kefei Tan

**Affiliations:** 1Qiqihar Branch of Heilongjiang Academy of Agricultural Sciences, Qiqihar 161006, China; mabo8210@haas.cn (B.M.); cjf69@163.com (C.L.); hujifang7@haas.cn (J.H.); zfyhhz@haas.cn (F.Z.); august-wjq@haas.cn (J.W.); 2Northeast Branch of National Saline–Alkali-Tolerant Rice Technology Innovation Center, Harbin 150000, China; liukailouis@163.com; 3Heilongjiang Academy of Agricultural Sciences, Harbin 150086, China; hljsdsgzh@haas.cn (Z.G.); songlijuan@haas.cn (L.S.); 4College of Computer and Control Engineering, Qiqihar University, Qiqihar 161006, China; zxxsnh@hrbeu.edu.cn

**Keywords:** saline–alkali-tolerant rice, Raman spectroscopy, Python, scipy.signal.filtfilt difference, identification feature information

## Abstract

Planting rice in saline–alkali land can effectively improve saline–alkali soil and increase grain yield, but traditional identification methods for saline–alkali-tolerant rice varieties require tedious and time-consuming field investigations based on growth indicators by rice breeders. In this study, the Python machine deep learning method was used to analyze the Raman molecular spectroscopy of rice and assist in feature attribution, in order to study a fast and efficient identification method of saline–alkali-tolerant rice varieties. A total of 156 Raman spectra of four rice varieties (two saline–alkali-tolerant rice varieties and two saline–alkali-sensitive rice varieties) were analyzed, and the wave crests were extracted by an improved signal filtering difference method and the feature information of the wave crest was automatically extracted by scipy.signal.find_peaks. Select K Best (SKB), Recursive Feature Elimination (RFE) and Select F Model (SFM) were used to select useful molecular features. Based on these feature selection methods, a Logistic Regression Model (LRM) and Random Forests Model (RFM) were established for discriminant analysis. The experimental results showed that the RFM identification model based on the RFE method reached a higher recognition rate of 89.36%. According to the identification results of RFM and the identification of feature attribution materials, amylum was the most significant substance in the identification of saline–alkali-tolerant rice varieties. Therefore, an intelligent method for the identification of saline–alkali-tolerant rice varieties based on Raman molecular spectroscopy is proposed.

## 1. Introduction

Saline–alkali fields are a widely distributed type of land presenting an agricultural barrier. They are often in uncultivated or semi-uncultivated land due to the poor suitability of saline–alkali soil for cultivation and the harm of saline–alkali to crops [1]. Rice is a moderately saline–alkali-tolerant plant and is considered the “pioneer” crop for soil improvement in saline–alkali land, which can accelerate the desalination and accumulation of organic matter by taking advantage of its unique advantages of saline–alkali resistance and growth in water [2,3]. Thus, planting rice in saline–alkali land can increase grain yield, improve saline–alkali soil and improve the ecological environment [4].

However, there are great differences in saline–alkali tolerance among different rice varieties [5], so rice breeding experts pay much attention to the identification and evaluation of saline–alkali-tolerant rice varieties. Wang et al. conducted a screening and identification of saline–alkali tolerance of japonica rice varieties in cold areas through an investigation and analysis of phenotypic morphological indicators [6,7,8]. However, this traditional identification method requires a large number of field tests, which are not only subject to the limitations of growth cycle and the natural environment, but also have the problems of a heavy workload and time consumption. Some researchers used genomics and proteomics methods to identify QTLs, analyzed the role of specific protein-linked genes, then identified and cultivated saline–alkali rice varieties by marker-assisted selection [9,10,11]. Genetic identification methods still require professionals to operate advanced equipment skillfully, and then analyze complex data, which is not only complicated in the detection process, but also high in cost.

Raman spectroscopy was first used in physical and chemical studies by Hibben J H and Teller E [12], which was mainly used to study the vibration and structure of molecular groups [13]. Since then, Raman spectroscopy technology has been widely applied in many other fields: food field [14], geological field [15], medical field [16] and agricultural field [17], etc. Giang Le Truong obtained sample information from 32 rice varieties in Vietnam through Raman spectroscopy, and conducted a multivariate analysis, such as PCA, KNN and HCA, to evaluate and identify rice varieties [18]. Japanese researchers established spectral information at the molecular level by Raman spectrometer on amylose, amylopectin, protein content and chemical residues of six rice varieties in Japan and proposed to use bar codes specially formulated for Raman spectroscopy to show the nutritional quality of labeled rice products [19,20]. In a word, a Raman spectrometer measurement has the advantages of fast, efficient and accurate, and Raman spectroscopy as a molecular spectrum can achieve rapid, accurate and nondestructive analysis of samples.

Python not only has the matrix manipulation functionality supported by commercial high-level languages such as Matlab, but it is also cleaner and more concise than languages such as Java and C [21]. Python has become the most popular programming language for big data analysts and has a powerful deep learning method that visualizes data from different neighborhoods to further analyze the visualizations [22,23,24]. Python can effectively deal with various interferences of spectral data based on its data visualization features and chooses the best algorithm for spectral data disturbance reduction according to the visualization effect. Python can extract spectral crests and features of the crest based on its data visualization features. Python can automatically reduce the dimensions of spectral features based on its powerful deep learning method function, which not only reduces the calculation time of the identification model, but also effectively improves the identification effect of the optimized model. The fingerprint identification of selected feature information is carried out according to the discriminant result of the optimized model.

At present, the application of Raman spectroscopy combined with Python in the identification of saline–alkali-tolerant rice varieties have not been reported. In view of this, this study used Raman spectrometer to acquire the molecular information of rice varieties, combined with Python deep learning and visual analysis method of data, was used to establish an identification model for saline–alkali-tolerant rice varieties [25]. The main purpose of this study is to obtain the identification model of saline–alkali-tolerant rice and provide an intelligent method for rice breeders to identify saline–alkali-tolerant rice varieties.

## 2. Materials and Methods

### 2.1. Test Material

The test materials were two saline–alkali-tolerant rice varieties and two saline–alkali-sensitive rice varieties [26,27], which were, respectively, taken from control soil and saline–alkali soil, and identified, screened and provided by Qiqihar Branch of Heilongjiang Academy of Agricultural Sciences. Soil basic nutrients and saline–alkali status was shown in Table 1.

Saline–alkali-tolerant coefficients (STC), namely, the relative values of each saline–alkali tolerance index, were used to evaluate the saline–alkali tolerance of rice, so as to eliminate the differences in basic traits among the tested materials [28]. The saline–alkali tolerance indexes in this experiment mainly investigated the four main agronomic traits of rice, namely plant height, tiller number, grain number per spike and 1000-grain weight.

The results showed that the STC of two saline–alkali-tolerant varieties BD6 and DF132 were significantly higher than those of the saline–alkali-sensitive varieties KD42 and LD12 in plant height, tiller number, grain number per spike and 1000-grain weight (*p* < 0.05; Figure 1).

### 2.2. Sample Processing

In October 2021, four rice varieties (10 caves for each variety, a total of 40 caves) obtained from the control field were dried naturally for 20 days in a laboratory of 22–25 °C, and the water content was reduced to about 15%. A total of 10 ears were taken from different positions in each cave, and 10 grains were taken from different positions of each ear, a total of 1000 grains. These grains were shelled by Shanghai Chao Xing LJJM for 40 s. 39 rice grains (Table 2) with complete appearance after shelling for each variety were selected as samples, and a total of 156 samples were obtained.

### 2.3. Obtaining Spectral Information

The sample information was collected by Advantage 532 Desktop Raman spectrometer. The excitation wavelength was 532 nm, which was an ideal light source for resonance Raman research, the measurement range was 200–3400 cm^−1^, the excitation power was less than 5 mw, the resolution was 1.4 cm^−1^, and the spectral information of 156 samples was obtained with 4 scanning times. Pro Scope HR software was used to obtain sample image information and sample data, which were saved in PRN format. The laser power was high, integration time was 4, number of spectrums was 3, display was average, save spectrum was ASCII and resolution was low. The data processing software was Python.

## 3. Results and Analysis

### 3.1. Disturbance Reduction and Crest Extraction

#### 3.1.1. Extraction of Original Raman Spectral Information

The raw data extracted from Python are shown in Figure 2. When using Raman spectrometer to collect rice spectral data, due to the interference of the instrument in terms of noise, stray light and fluorescence background, the data accuracy was affected. The spectral information of the four rice varieties was intertwined in a disorderly manner, making it difficult to distinguish. Therefore, it is essential to denoise and remove impurities from the original Raman spectral data.

#### 3.1.2. Reduction in Disturbance of Scipy.Signal.Lfilter Method

The scipy.signal.lfilter works with many basic data types when dealing with data dis-turbance reduction by Python. This filter is an implementation of the transpose of the di-rect form of the standard difference equation [29]. In this experiment, scipy.signal.lfilter was used, and the results are shown in Figure 3. The curve information of disturbance reduction by the scipy.signal.lfilter method was obviously smooth compared to that of the original data. This scipy.signal.lfilter method obtained a good effect of data disturbance reduction.

#### 3.1.3. Reduction in Disturbance of Scipy.Signal.Filtfilt Method

The scipy.signal.filtfilt method could quickly help to achieve a reduction in the interference of data. This method used a linear digital filter, which had twice the order of other filters, once forward and once back [30]. As shown in Figure 4, the curve processed by the scipy.signal.filtfilt method was smoother than that processed by scipy.signal.lfilter method. So, the scipy.signal.filtfilt method obtained better disturbance reduction effect on the original data of the Raman spectrum.

#### 3.1.4. Extraction of Wave Crest by Difference Method of Scipy.Signal.Filtfilt Method

Although the curve processed by scipy.signal.filtfilt had a good effect of removing disturbance, it filtered the useful spectral crest information. Therefore, the extraction of spectral crest information is essential. The improved signal filtering method, namely the scipy.signal.filtfilt method difference method, was used to extract the spectral crest information, as shown in Formula (1):

Differential equation:(1)X=x−y

Note: The x is the axis of data to be filtered; the y is the filtered data axis.

The full band of Raman spectrum was clearly visible, as shown in Figure 5. The eighteen significant crest information appeared in each rice variety. Referring to wave crest extraction, Raman spectral characteristics and the attribution of rice [31], seven effective crests were extracted at 480, 865, 941, 1129, 1339, 1461 and 2910 cm^−1^. Each rice variety had different wave intensities near the same wave value. Seven effective crests extracted by the signal filtering difference method laid the foundation for the extraction of crest feature in the next step.

### 3.2. Extraction of Wave Crest Feature

Four feature data for each wave crest were extracted by scipy.signal.find_peaks method in Python. As shown in Figure 5, four feature information (prominence, width, width_height and peak_dif) could accurately lock the shape and position of each wave crest.

The experiment consisted of 156 samples; each sample had seven crests and contained 4-dimensional feature information, so each sample owned 28-dimensional feature information. In addition, each sample owned its own three-dimensional feature information, i.e., name of sample, variety of sample and number of samples. Each sample owned 31-dimensional feature information based on the initial feature extraction. With the above machine learning approach, a 156 × 31-dimensional matrix feature information was established.

### 3.3. Reduction in Dimensionality of Features Information

If the feature information of 156 × 31-dimension matrix is directly brought into the classification model for machine deep learning, the large feature information matrix will lead to a large amount of computation and a long training time. Therefore, it is essential to reduce the dimensions of feature information.

#### 3.3.1. SKB Method for Dimensionality Reduction

The working principle of SKB is to use a certain parameter to score features and select the best several feature information. It is known as a large variable-characteristics selection tool. In this experiment, the SKB method selected mutual_info_regression parameters, and the mutual information method scored the 28-dimensional feature information of dimensionality unreduced, as shown in Formula (2) [32]. The 10 most powerful feature information were finally selected by the mutual trust method, as shown in Table 3.
(2)$$I\left(X;Y\right)=\sum_{x\in X}\sum_{y\in Y}p\left(x,y\right)\log\frac{p\left(x,y\right)}{p\left(x\right)p\left(y\right)}$$

Note: The mutual information method is used to evaluate the correlation between category independent variables and category dependent variables.

The SKB method selected seven crests and 10-dimensional feature information. Compared with the initial method, the select rate of effective crest was 100%, and the selection rate of effective feature information was 35.71%. A 156 × 13-dimensional matrix feature information was established.

#### 3.3.2. RFE Method for Dimensionality Reduction

The main idea of RFE is to iteratively build the feature of the model, eliminate the redundancy between features, select the optimal feature combination, and reduce the feature dimension. This experimental method used the Random Forest Classifier (RFC) module [33]. Firstly, the initial subset of 28-dimensional features was input into the RFC module, the importance of each feature was calculated, and the classification accuracy of the initial feature subset 1 was obtained by a cross-validation method. Second, the feature with the lowest significance was removed from feature subset 1, and feature subset 2 was obtained, which was input into the RFC module again to obtain the classification accuracy of subset 2. The above process was repeated until the feature subset is empty. Finally, several feature subsets with different number of features were obtained, and the feature subset with the highest classification accuracy was selected as the optimal feature combination. Therefore, this was a greedy algorithm to find the optimal feature subset, and finally selected 14 optimal feature data, as shown in Table 3.

The RFE method selected seven crests and 14-dimensional feature information. Compared with the initial method, the select rate of effective crest was 100%, and the selection rate of effective feature information was 50%. A 156 × 17-dimensional matrix feature information was established.

#### 3.3.3. SFM Method for Dimensionality Reduction

SFM is a built-in general transformer model in the feature selection method, which can perform feature selection through the indicators given by the model itself. In this experiment, the free feature selection method in SFM was selected, and the importance degree of different features was obtained after training with RFC model [34]. Features were selected according to the weight of importance, and 11 optimal feature data were finally selected, as shown in Table 3.

The SFM method selected seven peaks and 11-dimensional feature information. Compared with the initial method, the select rate of effective crest was 100%, and the selection rate of effective feature information was 39.29%. A 156 × 14-dimensional matrix feature information was established.

Three feature information selection methods were used to reduce the dimension of the initial feature information, which reduced the feature matrix and computing time. Whether the dimensionality reduction method of feature information could accurately identify saline–alkali-tolerant rice varieties required the classification model to evaluate the effectiveness of the feature information selection.

### 3.4. Establishment of Recognition Model

Based on one feature extraction and three feature selection methods, the feature information of a 156 × 31-dimensional matrix was established by the initial method, the feature information of a 156 × 13-dimensional matrix was established by the SKB method, the feature information of a 156 × 17-dimensional matrix was established by the RFE method and the feature information of a 156 × 14-dimensional matrix was established by the SFM method. The sample data of each rice variety were divided according to 7:3. A total 109 samples were divided into training sets, and 47 samples were divided into test sets. In order to find a fast, convenient, economical, reliable and accurate recognition model for saline–alkali-tolerant rice varieties, four matrix feature data of different dimensions were brought into the classification model, respectively, to conduct machine learning and evaluate the selection method of feature information.

#### 3.4.1. Establishment of Logistic Regression Model (LRM)

LRM is a classification model in machine learning [35], whose input function is the result of a linear regression, as shown in Formula (3):(3)$$h\left(w\right)={\;w}_{1}x_{1}+{\;w}_{2}x_{2}+w_{3}x_{3}\ldots +b$$

Input the LRM output results into the sigmoid function, as shown in Formula (4):(4)$$\;\mathrm{Sigmoid}\;\mathrm{function}:\;g\left(\theta^{T}x\right)=\frac{1}{1+{\;e}^{-\theta^{T}x}}$$

The sigmoid function output results were in the interval of [0,1], in which the default machine threshold was 0.5, and the classification results of the unoptimized LRM were obtained, as shown in Table 4. The matrix feature information of four different dimensions was brought into the unoptimized LRM for the recognition of saline–alkali-tolerant rice varieties, and the recognition rates were 80.85%, 74.47%, 76.60% and 80.85%, respectively.

#### 3.4.2. Establishment of Random Forests Model (RFM)

RFM establishes a forest in a random way, and there are many decision trees in the forest. There is no correlation between each decision tree in RFM. For the classification algorithm, when a new input sample enters, each decision tree in RFM is asked to make a judgment, respectively, to see which category this sample should belong to, and then see which category is selected the most, and predict this sample to be that category. The criterion parameter in this test was the gini coefficient, namely the decision tree of the CART category, as shown in Formulas (5) and (6).
(5)$$Gini\left(D\right)=\overset{\left|y\right|}{\underset{k=1}{\sum }}\underset{k^{\prime }\neq k}{\sum }p_{k}p_{k^{\prime }}=1-\overset{\left|y\right|}{\underset{k=1}{\sum }}p_{k}^{2}$$

Note: Two samples were randomly drawn from dataset D with the probability that their class labels were inconsistent. Therefore, the smaller the *Gini* (D) value, the higher the purity of the dataset D.
(6)$$Gini\_index\left(D,a\right)=\overset{V}{\underset{v=1}{\sum }}\frac{\left|D^{v}\right|}{\left|D\right|}Gini\left(D^{v}\right)$$

Note: The attribute that minimizes *Gini_index* (D,a) after division was selected as the optimal score attribute.

The CART could not only classify and regression, but also handled discrete and continuous attributes, and the classification results of the optimized RFM were obtained, as shown in Table 4. The matrix characteristic information of four different dimensions was brought into the optimized RFM for the recognition of saline–alkali-tolerant rice varieties, and the recognition rates were 80.85%, 82.98%, 89.36% and 85.11%, respectively. The results showed that the overall recognition rate of RFM was higher than those of LRM.

### 3.5. Attribution of Spectral Features Information

The main components of rice include starch, protein, fat, etc. [31]. Due to different content and structure, different spectral vibration information can be obtained (Table 5). In this experiment, the Raman spectrum of rice reached obvious Raman feature crests at 480, 865, 941, 1129, 1339, 1461 and 2910 cm^−1^. The spectral attribute of feature crest at 480 cm^−1^ was amylum, and the pattern of manifestation was skeleton vibration. The spectral attribute of the feature crest at 865 cm^−1^ was amylopectin and sugar ring, and the pattern of manifestation was the vibration of C-H deformation and C-O ring. The spectral attribute of the feature crest at 941 cm^−1^ was amylopectin, and the pattern of manifestation was symmetric stretching vibration of C-O-C. The spectral attribute of feature crest at 1129 cm^−1^ was sugar, and the pattern of manifestation was the vibration of C-O stretching and C-O-H bending deformation. The spectral attribute of feature crest at 1339 cm^−1^ was sugar, and the pattern of manifestation was C-O-H bending and the vibration of C-C stretching. The spectral attribute of feature crest at 1461 cm^−1^ was sugar, and the pattern of manifestation was C-H bending vibration in-plane. The spectral attribute of feature crest at 2910 cm^−1^ was amylum, and the pattern of manifestation was stretching vibration of CH_2_ and NH_2_.

## 4. Discussion

### 4.1. Interpretation of the Result of Reduction in Disturbance of Original Raman Spectral Information

With the improvements in the acquisition precision of Raman spectroscopy instruments, the collected Raman spectroscopy data contain terms of noise, stray light and fluorescence background [36]. These disturbances seriously affect the prediction accuracy of the model [37]. The effect of various interferences on model accuracy can be effectively eliminated by filtering disturbance reduction technology [38,39,40]. Python can visualize data from different neighborhoods to further analyze the visualization results [22,23,24]. In the Python language environment, scipy.signal.lfilter is a one-dimensional digital filter, and scipy.signal.filtfilt is called a zero-phase digital filter. In the basic implementation process of scipy.signal.filtfilt method, the signal is firstly filtered by scipy.signal.lfilter method, and then the signal is time-domain reversed and filtered by scipy.signal.lfilter method again. In this way, the phase is zero after two filters [41,42]. In this study, the curve processed by scipy.signal.filtfilt was smoother than that processed by scipy.signal.lfilter (Figure 2, Figure 3 and Figure 4); the waveform processed by scipy.signal.lfilter had offset, but the one processed by scipy.signal.filtfilt did not, so the scipy.signal.filtfilt could be used for reducing the disturbance of original Raman spectral information well [43,44,45].

### 4.2. Interpretation of the Result of Dimensionality Reduction

In recent years, with the improvement of data collection and storage capacity, data dimension increases exponentially along with samples and frequently appears in many scientific neighborhoods [46]. For problems of large scale or dimensionality, accuracy of estimation and computational cost are two top concerns [47]. SKB, RFE and SFM can achieve effective reduction in the dimensions of data through the powerful deep learning methods of Python [21,48,49,50]. In this test, the SKB method used the mutual_info_regression parameter for feature dimensionality reduction, and the RFE and SFM methods used the RFC module for feature dimensionality reduction. Compared with the 28-dimensional features extracted by the original method (Table 3), these three dimensionality reduction methods finally obtained 10-dimensional features, 14-dimensional features and 11-dimensional features, respectively. Therefore, three methods of feature dimension reduction were superior in reducing data dimensions and increasing efficiency [51].

### 4.3. Interpretation of the Establishment of Recognition Model

For one feature extraction and three feature selection methods (Table 4), modeling performance was ranked as RFE-RFM > SFM-RFM > SKB-RFM > Initial-RFM = Initial-LRM > SFM-LRM > RFE-LRM > SKB-LRM. The reasons for this are as follows: firstly, in this study, two models were used, one of which was a probabilistic model (LRM), and the other an important machine learning model (RFM) [52]. RFM are able to handle multi-dimensional and multi-variety data [53], and the results of this study showed that RFM owned greater ability to identify saline–alkali-tolerant rice varieties than LRM. Secondly, when the 28-dimensional features extracted in this experiment were put into LRM and RFM, respectively, RFM did not improve the recognition rate compared to LRM. After dimensionality reduction, the feature dimension for SKB, RFE and SFM were 10, 14 and 11, respectively. When they were brought into LRM and SFM, compared with LRM, the recognition rate of RFM increased by 8.51, 12.76 and 4.26, respectively. Feature extraction and feature selection (feature dimension reduction) are not completely separated in practical application; there are few studies on the combination of the two and further research is needed [54]. Finally, the purpose of this research was to build a saline–alkali rice variety recognition model using stable and efficient machine learning algorithms, the literature has confirmed the superior performance of RFM in classification evaluation [53,55]; the RFE method can eliminate the features of a low contribution rate without increasing model errors through repeated iterations to select the best features [33,56]. Therefore, RFE dimension reduction method was combined with the RFM in this study to improve the recognition rate of the identification model. The results showed that the RFE-RFM model of saline–alkali-tolerant rice varieties had the best recognition rate (Table 4).

### 4.4. Interpretation of Attribution of Spectral Features Information

This study found (Table 5) that the attributions of the Raman spectral crest in this experiment were amylum, amylopectin, sugar and sugar ring, and the pattern of manifestation were skeleton vibration, the vibration of C-H deformation, C-O ring, symmetric stretching vibration of C-O-C, the vibration of C-O stretching, C-O-H bending deformation, C-O-H bending and the vibration of C-C stretching, the stretching vibration of CH2, and NH2 and C-H in-plane bending vibrations. Amylum contained amylopectin, which was the aggregation of sugar molecules and the most common storage form of carbohydrates in cells [57]. In saline–alkali-tolerant and saline–alkali-sensitive rice varieties, the carbohydrate transportation to grain was closely related to the filling stage, and the dynamic accumulation of amylum played a decisive role [58]. Under the same planting environment, the dynamic accumulation capacity of amylum of saline–alkali-tolerant rice varieties ware higher than that of saline–alkali-sensitive rice varieties at the filling stage [59]. Therefore, the attributions of Raman spectral crest and its pattern of manifestation provided a biological theoretical basis for the identification of saline–alkali-tolerant rice varieties.

## 5. Conclusions

The present study exhibits the feasibility of Raman spectroscopy combined with Python machine deep learning method for the rapid and accurate identification of saline–alkali-tolerant rice. To improve detection accuracy and efficiency, the spectral data were preprocessed by the combination of scipy.signal.filtfilt and scipy.signal.find_peaks. Using the feature extraction combined with feature dimension reduction and classification modeling, the RFE-RFM models outperformed the other seven combined models. Therefore, it can quickly and accurately identify saline–alkali-tolerant rice varieties and provide variety guarantee for rice planting in saline–alkali fields.

## Figures and Tables

**Figure 1 plants-11-01210-f001:**
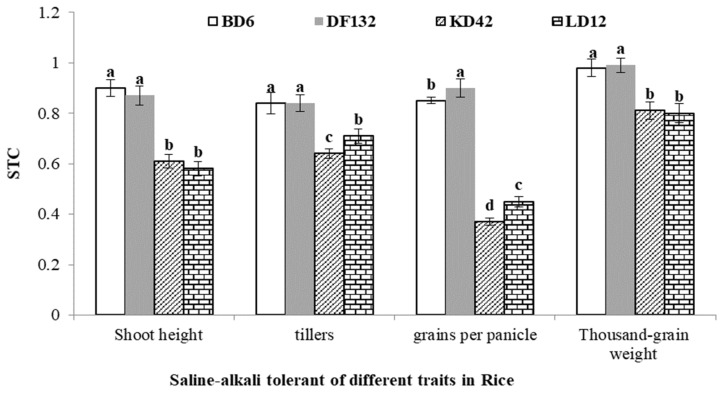
The saline–alkali tolerance coefficient of plant height, tiller number, grain number per ear and 1000-grain weight in rice. Values with different superscript letters are significantly different at *p* < 0.05. STC = index under saline–alkali stress/control index.

**Figure 2 plants-11-01210-f002:**
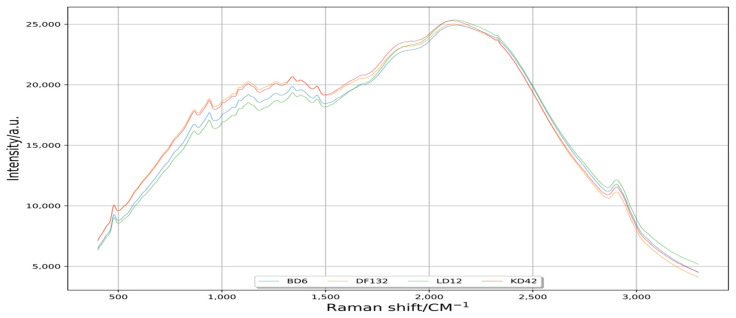
Original spectral curve. Raman shift is the reciprocal of wavelength, unrelated to the frequency of incident light and only related to the vibration frequency of sample molecules, and its range is 200–3400 cm^−1^. Intensity is the intensity of Raman scattering, which is the anti-Stokes line; the anti-Stokes line is the scattering light of frequency shifted light from monochromatic incident light in a molecule.

**Figure 3 plants-11-01210-f003:**
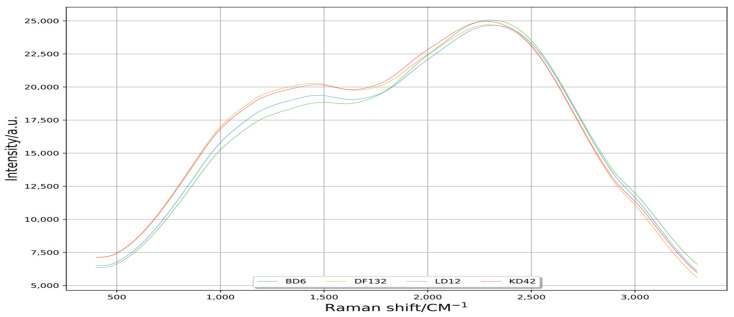
Spectral curves of disturbance reduction by filtering method. Raman shift is the reciprocal of wavelength, unrelated to the frequency of incident light and only related to the vibration frequency of sample molecules, and its range is 200–3400 cm^−1^. Intensity is the intensity of Raman scattering, which is the anti-Stokes line; the anti-Stokes line is the scattering light of frequency shifted light from monochromatic incident light in a molecule.

**Figure 4 plants-11-01210-f004:**
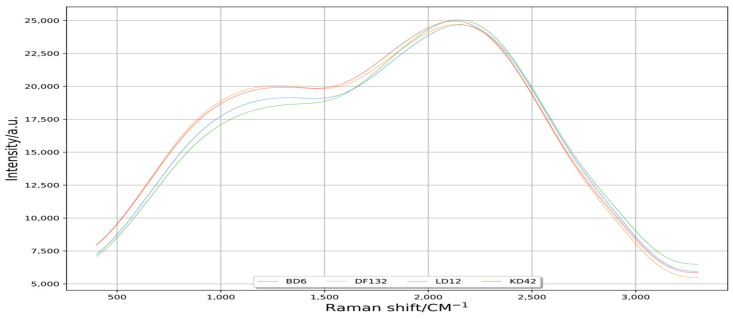
Spectral curves of disturbance reduction by signal filtering method. Raman shift is the reciprocal of wavelength, unrelated to the frequency of incident light, and only related to the vibration frequency of sample molecules, and its range is 200–3400 cm^−1^. Intensity is the intensity of Raman scattering, which is the anti-Stokes line; the anti-Stokes line is the scattering light of frequency shifted light from monochromatic incident light in a molecule.

**Figure 5 plants-11-01210-f005:**
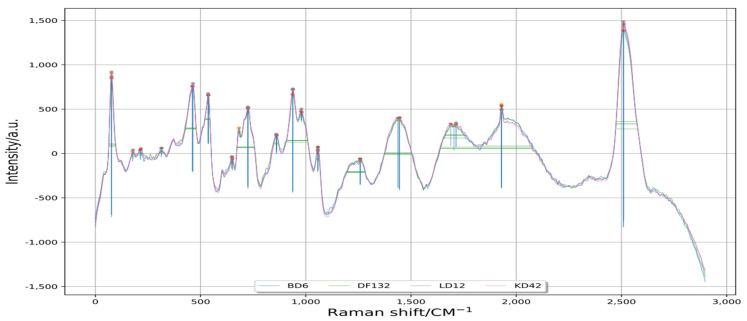
Crest and its four-dimensional feature information. Prominence is the length of the blue vertical line, width is the width of the green horizontal line, width_height is the length from the green line to the peak, and peak_dif is the peak offset. Raman shift is the reciprocal of wavelength, unrelated to the frequency of incident light and only related to the vibration frequency of sample molecules, and its range is 200–3400 cm^−1^. Intensity is the intensity of Raman scattering, which is the anti-Stokes line; the anti-Stokes line is the scattering light of frequency shifted light from monochromatic incident light in a molecule.

**Table 1 plants-11-01210-t001:** Basic nutrient status of soil (AHN: alkali-hydro nitrogen; AP: available phosphorus; RAP: rapidly available potassium; OM: organic matter).

Soil Types	AHN (mg/kg)	AP (mg/kg)	RAP (mg/kg)	OM (g/kg)	pH	Salt Content (%)
Saline–alkali soil Control soil	146.7 144.7	38.6 39.2	138.4 138.9	30.5 31.8	9.2 6.9	0.46 0.23

**Table 2 plants-11-01210-t002:** Variety and quantity of test samples (1: saline–alkali-tolerant; 0: saline–alkali-sensitive).

Numeral	Name of Sample	Variety of Sample	Number of Samples
1 2 3 4	BD6 DF132 KD42 LD12	1 1 0 0	39 39 39 39

**Table 3 plants-11-01210-t003:** Results of three methods for dimensionality reduction (p: prominences; w: width; wh: width_height; pd: peak_dif).

Number	Raman Shift/cm^−1^	Initial Feature Extraction	SKB	RFE	SFM
1 2 3 4 5 6 7	480 865 941 1129 1339 1461 2910	p\w\wh\pd p\w\wh\pd p\w\wh\pd p\w\wh\pd p\w\wh\pd p\w\wh\pd p\w\wh\pd	p\pd p\pd pd pd pd wh w\p	w\pd pd p\wh\pd pd p\wh\pd p\w\pd wh	pd pd pd pd p\wh\pd p\w\pd pd
Total		28	10	14	11

**Table 4 plants-11-01210-t004:** Modeling results of four feature selection methods for saline–alkali-tolerant rice varieties.

Feature Selection	Matrix Dimension	Accuracy		Accuracy Improvement
		LRM	RFM	
Initial SKB RFE SFM	156 × 31 156 × 13 156 × 17 156 × 14	80.85% 74.47% 76.60% 80.85%	80.85% 82.98% 89.36% 85.11%	0 8.51 12.76 4.26

**Table 5 plants-11-01210-t005:** Features and attribution of rice Raman spectra (s: strong; p: prominences; w: width; wh: width_height; pd: peak_dif).

Number	Raman Shift/cm^−1^	Pattern of Manifestation	Spectral Attribution	Methods	Feature Information			
1	480 s		amylum	SKB	p			pd
		Skeleton vibration		RFE		w		pd
				SFM				pd
2	865 s	The vibration of C-H	amylopectn sugar ring	SKB	p			pd
		deformation and C-O ring		RFE				pd
				SFM				pd
3	941 s	Symmetric stretching	amylopectn	SKB				pd
		vibration of C-O-C		RFE	p		wh	pd
				SFM				pd
4	1129 s	The vibration of C-O	sugar	SKB				pd
		stretching and C-O-H		RFE				pd
		bending deformation		SFM				pd
5	1339 s	C-O-H bending and	sugar	SKB				pd
		the vibration of		RFE	p		wh	pd
		C-C stretching		SFM	p		wh	pd
6	1461 s	C-H bending vibration	sugar	SKB			wh	
		in-plane		RFE	p	w		pd
				SFM	p	w		pd
7	2910 s	Stretching vibration of	amylum	SKB	p	w		
		CH_2_ and NH_2_		RFE			wh	
				SFM				pd
	Total				8	4	5	18

## Data Availability

The data and code presented in this study are openly available at github.com.

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
