# Peer review of "Intelligent Identification and Features Attribution of Saline–Alkali-Tolerant Rice Varieties Based on Raman Spectroscopy"

_plants, 2022, doi:10.3390/plants11091210_

Round 1
Reviewer 1 Report
Dear authors,
I had a great opportunity to review the research manuscript entitled "Identification and Features Attribution of Saline-alkali Toler- 2
ant Rice Varieties Based on Raman Spectroscopy".
From my point of view, the article can only be considered for publication in Industrial Crop Products if it is highly improved in particular in the discussion and conclusions. In fact, it needs a major revisions. Below I list several questions and comments about the manuscript that I believe will improve it.
*Title
Please improve the title of the article. you can make an attractive title
*Introduction
Please add the originality of your work? what's new? it's the first time...
Please clarify the objective of your work at the last of the introduction.
*Materials and methods
Sample processing : please explain how to prepare samples before using Raman spectroscopy. Do you use fresh leaves or dried leaves ?
Please give us more detais.
Results
Please give more information in legends in all figures
Please explain in figures axes what's intensity au and Raman shift/cm-
discussion and conclusions
I noticed that This part need completly improve and you should use more references and discuss more litterature.
Good luck
Reviewer 2 Report
The Authors have spent significant efforts in a novel approach to determine the saline-alkali tolerance of rice varieties by means off Raman sp@ectroscopy. The study is thoroughly conducted and deserves publication. However, some major points should be addressed, as follows:
1) The wavelength of the incoming laser is not shown. This is an important detail since the spectra are strongly affected by fluorescence. I believe that a more appropriate choice of laser wavelength could help to improve the spectral quality. The Authors should at least mention this point.
2) The Authors use Python procedures to treat the spectra, which is an approach with some novelty. However, the description is confusing and perhaps can be understood only by a quite limited number of experts of this technique. The explanation should be improved.
3) The interpretation of the spectrum at the molecular level is quite poor. They show amylopectin, but where is the amylose band? The definition of amylum at the molecular and spectroscopic level is unclear. Which vibrational mode of which molecule are they screening? A more detailed description of the spectrum at the molecular scale and a clearer description of the reason the found spectral difference links to the saline-alkali tolerance from a plant physiology viewpoint are needed.
In summary, the use of the Raman spectroscopic method is yet very empirical. However, since there is originality in this work, I will be happy to review the paper again after the above improvements are made.
Round 2
Reviewer 1 Report
Dear authors,
The manuscript has been correctly revised
Good luck
Reviewer 2 Report
The Authors have assessed all my comments and the paper is now suitable for publication.